# Clinical and Molecular Features of KRAS-Mutated Lung Cancer Patients Treated with Immune Checkpoint Inhibitors

**DOI:** 10.3390/cancers14194933

**Published:** 2022-10-08

**Authors:** Dan Zhao, Haiqing Li, Isa Mambetsariev, Tamara Mirzapoiazova, Chen Chen, Jeremy Fricke, Prakash Kulkarni, Victoria Villaflor, Leonidas Arvanitis, Stanley Hamilton, Michelle Afkhami, Raju Pillai, Brian Armstrong, Loretta Erhunmwunsee, Erminia Massarelli, Martin Sattler, Arya Amini, Ravi Salgia

**Affiliations:** 1Department of Medical Oncology and Therapeutics Research, City of Hope, Duarte, CA 91010, USA; 2Department of Gastrointestinal Medical Oncology, M.D. Anderson Cancer Center, The University of Texas, Houston, TX 77030, USA; 3Integrative Genome Core, Beckman Research Institute, City of Hope, Duarte, CA 91010, USA; 4Department of Computational & Quantitative Medicine, Beckman Research Institute, City of Hope, Duarte, CA 91010, USA; 5Department of Pathology, City of Hope, Duarte, CA 91010, USA; 6Light Microscopy/Digital Imaging Core, City of Hope, Duarte, CA 91010, USA; 7Department of Surgery, City of Hope, Duarte, CA 91010, USA; 8Department of Medical Oncology, Dana-Farber Cancer Institute, Boston, MA 02215, USA; 9Department of Medicine, Harvard Medical School, Boston, MA 02115, USA; 10Department of Radiation Oncology, City of Hope, Duarte, CA 91010, USA

**Keywords:** KRAS, immune checkpoint inhibitors, next-generation sequencing, precision medicine

## Abstract

**Simple Summary:**

The molecular and clinical features of KRAS-mutated lung cancer patients treated with immunotherapy have yet to be well characterized, and little information is known about resistance in these patients. The goal of this study is to better understand the survival results of KRAS-mutated patients who undergo immunotherapy treatment. For this effort, we have included 87 patients with NSCLC who received immunotherapy at the City of Hope, and we found that, among 87 patients, 32 had a KRAS G12C mutation (36.8%), 19 had G12V (21.9%), 18 had G12D (20.7%), 6 had G12A (6.9%), 3 had G12R (3.45%), and 10 had amplification (11.49%) and other uncommon mutations. G12D patients were found to respond differently compared to other KRAS-mutated patients. The OS with other KRAS comutations was not statistically significant, including STK11 and KEAP1. KRAS mutation subtypes such as G12D and comutations such as CDKN2/A and MET may modulate the immunotherapy responses and outcome in lung cancer.

**Abstract:**

Background: The molecular and clinical features of KRAS-mutated lung cancer patients treated with immunotherapy have yet to be characterized, which could guide the development of therapeutics targeting KRAS with potential immuno-oncology treatment combinations. Research Question: Do KRAS-mutated patients with different subtypes and comutations have different clinical responses and overall survival (OS) to checkpoint inhibitors? Study Design and Methods: 87 patients with NSCLC at the City of Hope who received immune checkpoint inhibitors were identified and analyzed retrospectively. Tumor genomic alterations were extracted from the clinical data with next-generation sequencing using various platforms. Demographic, clinical, molecular, and pathological information was collected with the approval of the institutional review board of the City of Hope. OS was calculated if it was available at the study time point, and responses were determined according to the RECIST v1.1. Results: Among 87 patients, 32 had a KRAS G12C mutation (36.8%), 19 had G12V (21.9%), 18 had G12D (20.7%), 6 had G12A (6.9%), 3 had G12R (3.45%), and 10 had amplification (11.49%) and other uncommon mutations. G12D had a statistically significant Odds Ratio (OR) between patients who had responses and progression of the disease (OR (95% CI) = 0.31 (0.09–0.95), *p* < 0.05), with 5 G12D-mutated patients having responses and 11 G12D-mutated patients having progression of the disease. In the univariate analysis with OS, there was a trend of better OS in the G12D-mutated patients, with no statistically significant difference in terms of OS between the patients who had G12D mutation and the patients who had other KRAS mutations (HR (95% CI) = 0.53 (0.21–1.36), *p* = 0.185). The median OS was significantly worse with KRAS comutation CDKN2A/B loss (4.2 vs. 16.9 months, HR = 3.07 (1.09–8.69), *p* < 0.05) and MET (3.4 vs. 17 months, HR = 3.80 (1.44–10.05), *p* < 0.01), which were included for the multivariate analysis. The OS with other KRAS comutations was not statistically significant, including STK11 and KEAP1. Conclusion: KRAS mutation subtypes such as G12D and comutations such as CDKN2/A and MET may modulate the immunotherapy responses and outcomes in lung cancer.

## 1. Introduction

Kirsten rat sarcoma viral oncogene homolog (KRAS) is the most common oncogenic driver in solid tumors including lung cancer and was associated with a worse prognosis and resistance to chemotherapy and anti-epidermal growth factor receptor (EGFR) treatment [1,2]. Targeting KRAS has been challenging for decades due to the lack of known drug-binding pockets. Recently, allosteric KRAS^G12C^ mutant-specific inhibitors that covalent bind to the mutant cysteine beneath the switch-II region, which locks it at the inactive GDP bound form, were discovered [3,4]. Early phase clinical trials of the KRAS^G12C^ inhibitors including sotorasib (AMG510) and adagrasib (MRTX849) in solid tumors were encouraging, and Sotorasib was FDA-approved for previously treated KRAS^G12C^-mutated non-small cell lung cancer (NSCLC) [5,6,7,8,9,10,11]. The first-in-human phase 1 trial of Sotorasib showed a disease control rate of 90% (5 partial responses, 4 stable diseases) in the 10 NSCLC patients, and the phase 2 trial reported a 37.1% response rate and an 80.6% disease control rate with a median duration of response of 11.1 months in previously treated KRAS^G12C^-mutated advanced lung cancer patients [7,12]. Interestingly, treatment with the KRAS^G12C^ inhibitor resulted in a pro-inflammatory tumor microenvironment and synergistic effects with immunotherapy with increased T cells, macrophages, and dendritic cells infiltration [5]. The RAS/MAPK pathway is essential for T cell development, proliferation, differentiation, and function [13]. Challenges for targeting KRAS include the limited response rate and the short duration of the response to KRAS inhibitors, which prompted the early clinical investigation of combining KRAS-targeted therapy with immunotherapy including immune checkpoint inhibitors (ICIs) in KRAS-mutated patients [14].

Immune checkpoint inhibitors (ICIs) are currently used as monotherapy or combination therapy in the frontline and subsequent lines for metastatic non-small cell lung cancer (NSCLC) [15,16,17,18,19]. Furthermore, ICIs before or after surgery showed efficacy in patients with resectable disease in neoadjuvant and adjuvant settings, highlighting the potential of ICIs to improve outcomes in this patient group [20,21]. The response rates in lung cancer to ICIs are approximately 20% for monotherapy and 40% for combination therapy, but eventually, most patients have progression of the disease, and overcoming resistance is an unmet need [22,23]. Mutated KRAS causes phenotypic switching of naïve T cells to immune suppressive Treg-like cells, possibly with metabolic changes of less utilization of glucose-6-phosphate [24]. Targeting KRAS could potentially overcome the primary and acquired resistance to immunotherapy. Multiple trials are currently ongoing, combining KRAS^G12C^ inhibitors with ICIs in cancer patients.

KRAS-mutated cancers are heterogeneous, and genomic commutations, MET amplification, metabolic reprogramming, and EGFR signaling represent some possible mechanisms of resistance to the KRAS^G12C^ inhibitors and immunotherapy [6,25,26,27,28,29]. The predictive role of KRAS mutation in checkpoint inhibitors’ treatment outcomes is inconsistent with their genetic heterogeneity and complexity [30,31]. KRAS-mutated patients benefit from immunotherapy, and KRAS mutations were not found to be different in the overall population compared with patients who had durable clinical benefits with checkpoint inhibitors [32,33,34,35,36]. The comutations with KRAS were reported to be the primary drivers of molecular and immunological differences in KRAS-mutant lung adenocarcinomas, while specific KRAS mutations (KRAS^G12C^, KRAS^G12V^, KRAS^G12D^, and others) did not have a consistent pattern [37]. The comutation of STK11/LKB1 was reported to have higher KEAP1 mutational inactivation and fewer immune cells, while the comutation of TP53 was associated with higher inflammatory markers and longer relapsed free survival in KRAS-mutated lung cancer [37]. In an in vitro study, the genomic loss of KEAP1 represented a mechanism of resistance to the KRAS^G12C^ inhibitor adagrasib (MRTX849) [6]. The comutation of STK11/LKB1 and the comutation of KEAP1/NFE2L2 were identified as genomic drivers for primary resistance to immune checkpoint inhibitors in KRAS-mutated lung adenocarcinoma [38,39].

With the development of KRAS inhibitors and combination strategies of KRAS inhibition with immunotherapy and other targeted therapies, there is an unmet need of characterizing the clinical and molecular features of KRAS-mutated lung cancer patients treated with immunotherapy to facilitate pre-clinical investigations and clinical development. Clinical and molecular profiling is needed for the selection of patients, and the identification of novel therapeutic targets and strategies to improve the response rates and the duration of responses to treatments including ICIs and KRAS inhibition in KRAS-mutated patients. In this study, we analyzed the clinical and molecular characteristics of 87 lung cancer patients with mutated KRAS who had received monotherapy ICIs to identify the associations with the clinical outcomes of responses and overall survival (OS).

## 2. Patients & Methods

### 2.1. Patients

Patients (*n* = 87) with NSCLC at the City of Hope National Medical Center who received ICIs (pembrolizumab, nivolumab, atezolizumab, or durvalumab) were identified retrospectively in different settings, with the cutoff date of 11 August 2018, including standard of care, compassionate use, and clinical trials. The information on tumor genomic alterations (GAs) was extracted from the available clinical data, including mutations in KRAS, EGFR, TP53, and PD-L1. The testing using various next-generation sequencing (NGS) platforms included FoundationOne (Foundation Medicine, Cambridge, MA, USA), Caris (Caris Life Science, Phoenix, AZ, USA), Paradigm (Paradigm Diagnostics, Phoenix, AZ, USA), Guardant360 (Guardant, Redwood City, CA, USA), NeoGenomics (NeoGenomics Laboratories, Fort Myers, FL, USA), or targeted gene sequencing panels at the City of Hope. The Tumor Proportion Score (TPS) was used to quantify PD-L1 (22C3) expression by IHC. Negative PD-L1 is defined as <1% of viable tumor cells showing membranous staining.

The City of Hope institutional review board approved the collection of demographic, clinical, and pathological information. The informed consent was waived, as per the IRB guidelines for retrospective studies on clinical and molecular information. Overall survival (OS, from the start of the ICIs) was calculated if it was available at the study time point. Responses were determined by clinical and radiological evaluation according to the RECIST v1.1 criteria [40].

### 2.2. Statistical Analysis

The OS was defined as the overall survival from the start of ICI treatment until death. In this study, the Hazard Ratios (HR) are estimated using overall survival (OS). The univariate COX proportional hazards model was used to test the association of clinical and molecular features with OS independently first. Based on the univariate analysis result, clinically and biologically relevant features with statistical significance (cutoff *p*-value < 0.05) were selected for the multivariate COX proportional hazards model analysis. PD-L1 expression was categorized as negative (<1%), 1%–<50%, and ≥50%. Overall survival (OS) was estimated using the Kaplan–Meier method. The difference in survival curves was tested using the Log-rank test. GraphPad Prism 8 (GraphPad Software) and R ver. 3.6.2 were used for the statistical analyses and data visualization. All tests were two-sided, and statistical significance was identified with a *p*-value < 0.05.

## 3. Results

### 3.1. Patient Characteristics

The baseline characteristics of 87 KRAS-mutated patients were summarized based on their age, sex, smoking status, histology, TP53 status, and PD-L1 expression (Table 1). The median age was 68.5 years (range 49–89). A total of 38 (43.7%) patients were ≥70 years old and 49 (56.3%) were <70 years old at the beginning of the ICI treatment. A total of 42 (48.3%) were female and 45 (51.7%) were male; 16 (18.4%) were never smokers, 61 (70.1%) were former smokers, and 10 (11.5%) were current smokers. The histology was predominantly adenocarcinoma (*n* = 83, 95.4%), with two cases of squamous cell lung cancer (2.3%) and two (2.3%) cases of other types (one poorly differentiated large cell lung cancer and one poorly differentiated carcinoma). PD-L1 was tested in 66 patients: 21 (32.3%) were negative (<1%), 31 (47.7%) were ≥50%, and 13 (20%) were between 1% and <50%. TP53 was tested in 79 patients—38 (48.1%) positive and 41 (51.9%) negative patients. A total of 84 patients (96%) had stage IV disease, while 2 patients had Stage IIIB disease and 1 patient had stage IIIA disease at the time of diagnosis.

### 3.2. KRAS Mutation Subtypes with Responses and OS

Among 87 patients, 32 of them had KRAS^G12C^ (36.8%), 19 had KRAS^G12V^ (21.9%), 18 had KRAS^G12D^ (20.7%), 6 had KRAS^G12A^ (6.9%), 3 had KRAS^G12R^ (3.45%), 10 had KRAS amplification (11.49%), 1 had KRAS^G12S^ (1.1%), 1 had KRAS^G13D^ (1.15%), 1 had KRAS^G13R^ (1.15%), 2 had KRAS^Q61L^ (2.3%), 2 had KRAS^Q61H^ (2.3%), 1 had KRAS^K117N^ (1.15%), and 1 had all three of KRAS^G12D^, KRAS^G12V^, and KRAS^G12R^ (1.15%) (Figure 1). G12D has a statistically significant Odds Ratio (OR) between patients that had responses and progression of the disease (OR (95% CI) = 0.31 (0.09–0.95), *p* < 0.05) and 5 KRAS G12D-mutated patients who had responses and 11 KRAS G12D-mutated patients who had progression of the disease. In the univariate analysis with OS, there was a trend of a better OS in KRAS G12D-mutated patients, with no statistically significant difference in OS between patients who had G12D mutations and patients who had other KRAS mutations (Table 2, HR (95% CI) = 0.53 (0.21–1.36), *p* = 0.185). However, in the multivariate analysis, G12D-mutated patients had a longer OS compared with patients who had other KRAS genomic alterations (Table 3, HR (95% CI) = 0.09 (0.01–0.68), *p* = 0.02). Patients who had KRAS G12V mutations had a trend of a worse OS that was not statistically significant by either univariate (Table 2, HR (95% CI) = 1.94 (0.95–3.96), *p* = 0.068) or multivariate analysis (Table 3, HR (95% CI) = 4.13 (0.98–17.50), *p* = 0.053). For the other KRAS mutation subtypes, we did not find statistical significance with OS nor with responses.

### 3.3. KRAS Comutations and OS

The top detected GAs and the patient’s clinical information (Figure 2) were sorted by the detected positive rate of GAs among tested patients (the number of tested patients for each gene varied due to different gene panels in the testing platforms). TP53 (*n* = 38) ranked as the most frequently detected comutation with KRAS (87 patients), with a 48% positive rate in the 79 patients tested for TP53, followed by LRP1B (11/40, 28%), SPTA1 (9/34, 26%), ARID1B (9/35, 26%), SMARCA4 (11/43. 26%), MLL3 (9/37, 24%), STK11 (15/75, 20%), EPHA3 (7/37, 19%), KEAP1 (7/41, 17%), NKX2-1 (6/36, 17%), MLL (6/40, 15%), FAT1 (5/37, 14%), ATM (10/75, 13%), and NF1 (5/40, 13%).

The results showed CDKN2A/B loss (6/49, 12%), TSC2 (6/52, 12%), TET2 (4/37, 11%), and ARID2 (4/38, 11%). Some KRAS-mutated patients had targetable genomic alterations (GA). A total of 9% of the patients were positive for MET genomic alteration (7/79), including two amplifications, one exon 14 deletion, one T1010I mutation, one D428G mutation, one A347T mutation (Variants of Unknown Significance, VUS), and one T948H mutation (VUS). GAs of the EGFR gene were found in 5% of the patients (4/85: 1 patient had E282K mutation, 1 patient had T790M and L858R mutation, and 2 patients had amplification). In total, 4% had ERBB2 mutation (3/78), including two patients who had amplification and one who had E645K mutation. A total of 3% of the patients had BRAF amplification (2/81), and 1% of patients (1/80) were positive for ALK.

The univariate COX analysis revealed a statistically significant (*p* < 0.05) association between OS and PD-L1 status. Patients with a PD-L1 level ≥50% showed a longer OS compared with PD-L1-negative patients (HR; 95% CI; *p* < 0.01). The median OS for PD-L1-negative patients is 7.1 months. The median OS for patients who had PD-L1 between 1% and 50% is 8.1 months. For patients who had PD-L1 ≥ 50%, the lower 95% CI of the median OS is 19 months (Figure 3A). The median OS was also significantly worse with KRAS comutation CDKN2A/B loss (4.2 vs. 16.9 months, HR = 3.07 (1.09–8.69), *p* < 0.05, Figure 3B) and MET (3.4 vs. 17 months, HR = 3.80 (1.44–10.05), *p* < 0.01, Figure 3C) which were included for the multivariate analysis. The association of OS with other KRAS comutations was not statistically significant, including STK11 (HR = 1.13 (0.43–2.97), *p* = 0.812), KEAP1 (HR = 1.83 (0.51–6.54), *p* = 0.35), ARID1A (HR = 0.24 (0.03–1.80), *p* = 0.166), ATM (HR = 0.52 (0.16–1.72), *p* = 0.285), TSC2 (HR = 2.37 (0.66–8.46), *p* = 0.183), MYC (HR = 2.78 (0.91–8.43), *p* = 0.071), and others (Appendix A).

In the multivariate COX proportional hazards model analysis, we included PD-L1 status, genomic alterations (CDKN2A/B and MET) associated with OS by univariate analysis, and KRAS subtypes (G12D and G12V). We also included the demographic factors including gender, age group, and smoking status in the multivariate model. The multivariate analysis showed that PD-L1 remains statistically significant (HR (95% CI) = 0.12 (0.03–0.57), *p* < 0.01), as well as CDKN2A/B loss (HR (95% CI) = 9.44 (1.90–46.93), *p* = 0.006). The association of a worse OS with the KRAS comutation of MET (HR = 3.80 (1.44–10.05); *p* < 0.01; Figure 3C) was not retained in the multivariate COX proportional hazards model (Table 3, HR = 3.46 (0.55–21.91), *p* = 0.186). No statistical significance was found for age, gender, and smoking status with OS in the univariate analysis and multivariate analysis.

## 4. Discussion

In this study, we analyzed the clinical and molecular features of 87 KRAS-mutated lung cancer patients treated with ICIs at the City of Hope. We characterized the KRAS mutation subtypes and comutations with responses to ICIs and survival outcomes. As expected, the higher PD-L1 expression level is associated with a longer survival. The median OS for negative PD-L1 expression (<1%), 1%–<50%, and ≥50% is 7.1 months, 8.1 months, and more than 19 months (*p* < 0.01).

KRAS comutation with CDKN2A/B loss was associated with a worse OS (median 4.2 vs. 16.9 months, HR = 3.07 (1.09–8.69), *p* < 0.05, Figure 3B). This is consistent with our previous findings of the least favorable outcome of lung cancer patients harboring CDKN2A/B loss treated with ICIs in another cohort with both KRAS-mutated and KRAS wildtype patients [41]. CDKN2A/B loss might have negative prognostic and predictive value for lung cancer with immunotherapy, as reported by others as well [42]. It was reported that cell cycle, SHP2, MYC, and mTOR were among the key pathways for cell fitness by CRISPR/Cas 9 knockout screening with the KRAS G12C inhibitor MRTX849, and the combination of the KRAS G12C inhibitor with the CDK4/6 inhibitor palbociclib demonstrated more tumor regression in xenograft tumor models than either single agent alone [6]. The genomic alteration of CDKN2A/B could be a predictive marker for the combination therapy of the KRAS inhibitor and the CDK 4/6 inhibitor in selected patients [6].

KRAS comutation with MET genomic alterations also resulted in a shorter median OS (3.4 vs. 17 months, HR = 3.80 (1.44–10.05), *p* < 0.01, Figure 3C). The MET receptor tyrosine kinase (RTK) and its ligand hepatocyte growth factor (HGF) play an important role in cancer development as well as in innate and acquired resistance to lung cancer treatment, including EGFR inhibition [43]. It has been well documented that MET genomic alterations were associated with worse outcomes of lung cancer immunotherapy [44,45]. MET bidirectionally regulates both cancer cells and different immune cells, and MET expression in monocytes/macrophages/neutrophils was associated with IL-10 expression and immunosuppressive myeloid cells [46,47,48,49]. The crosstalk of the MET and KRAS pathways could confer resistance to lung cancer-targeted therapies [50,51]. KRAS amplification contributes to the resistance to MET inhibition in lung cancer, and MET amplification was noticed in the acquired resistance to the KRAS G12C inhibitor [29,51]. The combination strategies of KRAS and MET targeting as well as immunotherapy warrant more investigation.

Consistent with previous findings in lung cancer, KRAS^G12C^ (32/87, 36.8%) was the most common mutation subtype in our study population, followed by G12V (19/87, 21.9%), G12D (18/87, 20.7%), G12A (6/87, 6.9%), and other mutations (Table 2, Figure 1 and [52]). There was more progression of the disease than the response in KRAS^G12D^-mutated patients (Table 2, OR (95% CI) = 0.31 (0.09–0.95), *p* = 0.048); however, the association of a better OS with KRAS G12D mutation was revealed in the multivariate analysis, which indicated that KRAS G12D-mutated patients, especially those who had responses with immunotherapy, had a longer OS compared with patients who had other KRAS mutations (Table 3, HR (95% CI) = 0.09 (0.01–0.68), *p* = 0.02). KRAS^G12V^ mutations had a trend toward a worse OS but were not statistically significant (Table 2 and Table 3). Due to the limited sample size, our study included only one patient with a KRAS^Q61L^ mutation, who had progression of the disease. The samples sizes were also not sufficient to detect the statistical significance in other KRAS mutation subtypes, such as other uncommon codon 12, codon 13, and codon 61 mutations.

There was heterogeneity in the prognostic and predictive roles of KRAS mutation subtypes with immunotherapy. Each KRAS mutation subtype has its unique biochemical and clinicopathological features, and the differences between the mutation subtypes in cancer and treatment are not fully understood yet [53,54,55,56]. The KRAS^G12D^ mutant has an intrinsic wild-type and SOS1 guanine exchange activities, while KRAS^Q61^ mutants were defected in GTP hydrolysis [55,57]. In another study of 144 KRAS-mutated non-squamous NSCLC patients, patients who had KRAS^G12C^ or other KRAS mutations had no significant difference in clinical features, treatment, and survival [58]. In a study with 218 KRAS-mutated Japanese patients treated with ICIs after first-line chemotherapy, KRAS^G12C^ was significantly associated with high TMB (≥10 mut/Mb), and KRAS^G12C^ or KRAS^G12V^ were associated with high PD-L1 expression (≥50%). The median progression-free survival (mPFS) was significantly longer in patients with KRAS G12C or G12V than it was in other KRAS mutations [59]. In total, 15% of the patients had STK11 mutations without a difference in the comutation frequency among the different KRAS mutation subtypes, and the mPFS was significantly shorter with STK11 comutations with KRAS G12C or G12V (1.8 vs. 5.7 months, HR 1.97 (95% CI 1.06–3.41), *p* = 0.02) [59]. In another cohort of 1194 patients with KRAS-mutated NSCLC treated at the Memorial Sloan Kettering Cancer Center, KRAS^G12C^ and other KRAS mutations (15% G12D, 16% G12V, 8% G12A, 4% G13D) had similar comutation patterns and outcomes with similar response rates between patients with KRAS^G12C^ or other KRAS mutations in patients with PD-L1 higher than 50% (*n* = 103, 40% vs. 58%, *p* = 0.06) [60]. While most of the previously published data were comparing G12C and non-G12C mutations, our findings on the association of G12D mutations with immunotherapy outcomes indicated further investigation into all allele-specific alterations or KRAS amplifications and their association with immunotherapy outcomes.

## 5. Conclusions

The associations of responses and OS with KRAS mutation subtypes and comutations were analyzed in 87 KRAS-mutated lung cancer patients treated with ICIs. We found that the comutation of MET and CDKN2A/B loss was associated with a worse OS. The limitations of this study are that it is a single-institution retrospective study with a limited sample size, and no correlative tissue and blood samples were included in this project. Our findings of different outcomes of KRAS G12D, G12V, and Q61L with immunotherapy warrant independent and larger population validation.

## Figures and Tables

**Figure 1 cancers-14-04933-f001:**
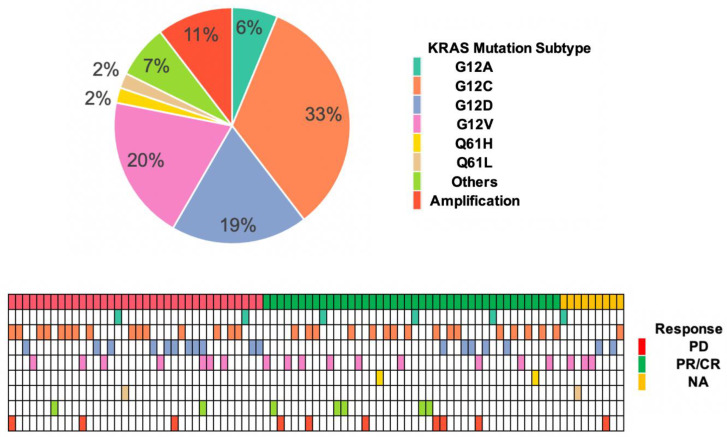
KRAS mutation subtype characteristics by response (*n* = 87).

**Figure 2 cancers-14-04933-f002:**
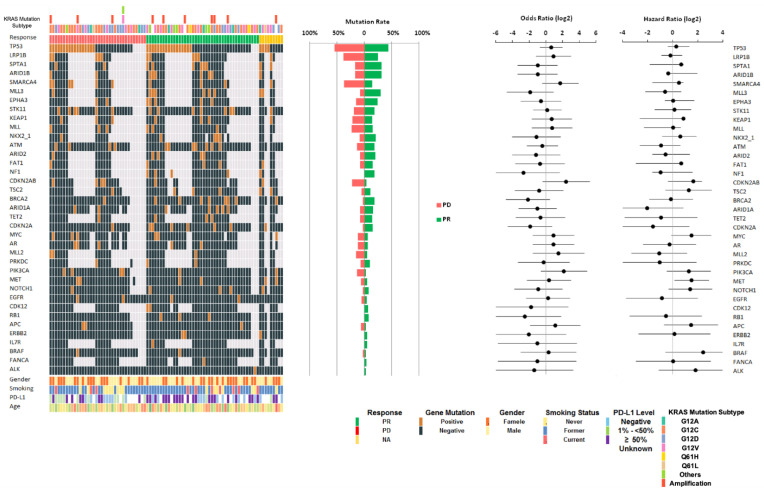
Oncoplot for KRAS common genetic co-occurring mutations (gray signifies that the gene was not tested) and their associations with response (odds ratio) and overall survival (hazard ratio).

**Figure 3 cancers-14-04933-f003:**
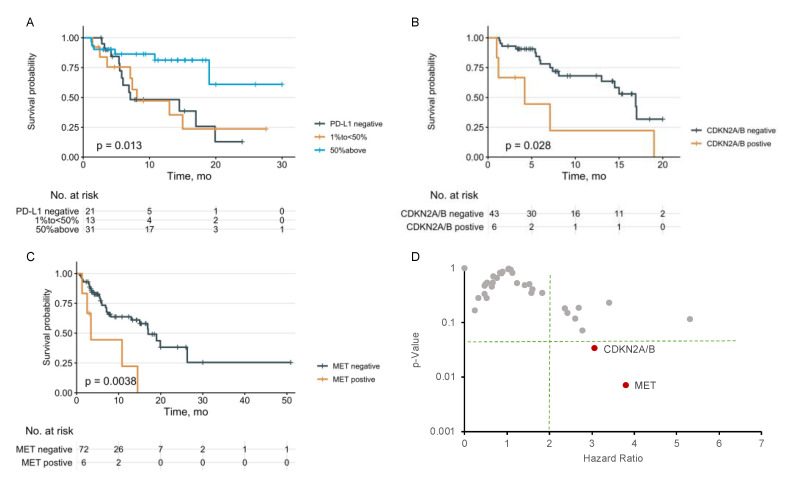
KRAS comutations with overall survival (OS) and Hazard Ratio. (**A**) OS with PD-L1 status in all KRAS-mutated patients tested (*n* = 65). (**B**) OS with CDKN2A/B comutation status in all KRAS-mutated patients tested (*n* = 49). (**C**) OS with MET comutation status in all KRAS-mutated patients tested (*n* = 78). (**D**) KRAS comutations with Hazard Ratio.

**Table 1 cancers-14-04933-t001:** Baseline patient characteristics (*n* = 87).

Characteristics	No. of Patients (*n* = 87)	CR/PR (*n* = 42)	PD (*n* = 36)	*p*-Value ^b^
Age, years, at ICIs				
<70	49 (56.3%)	24 (57%)	20 (56%)	1
≥70	38 (43.7%)	18 (43%)	16 (44%)	
Sex				
Women	42 (48.3%)	20 (47%)	18 (50%)	0.82
Men	45 (51.7%)	22 (53%)	18 (50%)	
Smoking status				
Current	10 (11.5%)	7 (17%)	3 (8%)	0.33
Former	61 (70.1%)	29 (69%)	24 (67%)	
Never	16 (18.4%)	6 (14%)	9 (25%)	
Histology				
Lung adenocarcinoma	83 (95.4%)	40 (95.2%)	34 (94%)	0.99
Lung squamous	2 (2.3%)	1 (2.4%)	1 (3%)	
Others ^a^	2 (2.3%)	1 (2.4%)	1 (3%)	
TP53				
Positive	38 (48.1%) ^c^	17 (44%)	17 (55%)	0.49
Negative	41 (51.9%) ^c^	22 (56%)	14 (45%)	
Total tested	79	39	31	
PD-L1				
Negative	21 (32.3%) ^c^	11 (32%)	8 (30%)	0.001
1–<50%	13 (20%) ^c^	1 (6%)	11 (40%)	
≥50%	31 (47.7%) ^c^	21 (62%)	8 (30%)	
Total tested	65	33	27	

^a^ Others included one poorly differentiated large cell carcinoma and one poorly differentiated carcinoma. ^b^
*p*-values derived from Fisher’s exact tests. ^c^ % Based on total tested patients.

**Table 2 cancers-14-04933-t002:** KRAS mutation subtypes with responses and OS (*n* = 87).

KRAS GAs	No. (*n* = 87)	CR/PR (*n* = 42)	PD (*n* = 36)	OR (95% CI)	OR *p*-Value	OS Association HR (95% CI)	HR *p*-Value
G12C	32 (36.8%)	16	15	0.86 (0.35–2.15)	0.748	1.00 (0.52–1.93)	0.997
G12V	19 (21.9%)	9	7	1.13 (0.37–3.53)	0.829	1.94 (0.95–3.96)	0.068
G12D	18 (20.7%)	5	11	0.31 (0.09–0.95)	0.048 *	0.53 (0.21–1.36)	0.185
G12A	6 (6.9%)	3	2	1.31 (0.21–10.37)	0.776	1.09 (0.38–3.09)	0.875
G12R	3 (3.5%)	1	1	0.85 (0.03–22.12)	0.912	0 (0–inf)	0.997
Q61H	2 (2.3%)	2	0	NA	0.992	0.34 (0.04–2.67)	0.306
Q61L ^a^	2 (2.3%)	0	1	NA	0.991	7.75 (1.71–35)	0.008 **
G12S	1 (1.1%)	1	0	NA	0.992	0 (0–inf)	0.997
G13D	1 (1.1%)	1	0	NA	0.992	0 (0–inf)	0.996
G13R	1 (1.1%)	0	1	NA	0.991	0 (0–inf)	0.997
K117N	1 (1.1%)	1	0	NA	0.992	2.41 (0.33–17.82)	0.338
KRAS ^a^ amp	10 (11.5%)	6	3	1.83 (0.45–9.24)	0.417	0.59 (0.18–1.92)	0.381
Other ^b^	1 (1.1%)	0	1	NA	0.991	0 (0–inf)	0.997

^a^ One patient had an unknown response status. ^b^ One patient had G12D, G12V, and G12R. * *p* < 0.05, ** *p* < 0.01.

**Table 3 cancers-14-04933-t003:** Multivariate analysis for OS.

Risk Factors	HR (95% CI)	*p*-Values ^1^
PD-L1		
Negative or less than 50%	Reference	
50% above	0.12 (0.03–0.57)	0.007 **
CDKN2A/B Loss		
Negative	Reference	
Positive	9.44 (1.90–46.93)	0.006 **
MET Mutation		
Negative	Reference	
Positive	3.46 (0.55–21.91)	0.186
KRAS G12V Mutation		
Negative	Reference	
Positive	4.13 (0.98–17.50)	0.053
KRAS G12D Mutation		
Negative	Reference	
Positive	0.09 (0.01–0.68)	0.02 *
Age		
<70	Reference	
>=70	1.26 (0.35–5.25)	0.66
Sex		
Female	Reference	
Male	1.53 (0.45–5.14)	0.49
Smoking Status		
Never	Reference	
Current	0.38 (0.04–3.25)	0.38
Former	0.33 (0.07–1.47)	0.15

^1^ Multivariate COX proportional hazards model for OS. * *p* < 0.05; ** *p* < 0.01.

## Data Availability

The data presented in this study are available in the manuscript and the Appendix A.

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
