# Peer review of "Clinical and Molecular Features of KRAS-Mutated Lung Cancer Patients Treated with Immune Checkpoint Inhibitors"

_cancers, 2022, doi:10.3390/cancers14194933_

Round 1
Reviewer 1 Report
The authors present an interesting study assessing the correlation between the specific KRAS mutations and therapeutic outcomes post immunotherapies. Their findings suggest an increased OS post-ICI treatment in patients harboring KRAS G12D mutation.
I have some minor comments:
1. Please include which immune-check point inhibitors were administered to the patients. Did all of them receive PD-1?
2. Please include limitations of this study
Reviewer 2 Report
The Manuscript is reporting a retrospective study to understand the effect of mutations in KRAS and co mutations on success of ICIs as treatment for lung cancers as measured by Overall survival. The study is performed well, I have following suggestions to improve the article.
1. The introduction should emphasize a bit more on the use of OS and HR. Hazard ratio is particularly not explained well.
2. The title of the article is a bit misleading as there are no experimental characterizations performed. It is a retrospective study and the title should reflect that.
3. There are some grammar edits needed:
for example:
The sentence for question in abstract should be rewritten to make it clear.
The titles in the figure 1 and figure 2 has typo: it writes "subtypes" as "subtytes" at multiple places in both figures.
